Protein signatures using electrostatic molecular surfaces in harmonic space

Carvalho C. Sofia 1 2 cscarvalho@oal.ul.pt
Vlachakis Dimitrios 3
Tsiliki Georgia 3
Megalooikonomou Vasileios 4
Kossida Sophia 3 skossida@bioacademy.gr
1 Centro de Astronomia e Astrofísica da Universidade de Lisboa , Tapada da Ajuda, Lisbon , Portugal
2 Research Center for Astronomy and Applied Mathematics, Academy of Athens , Athens , Greece
3 Bioinformatics & Medical Informatics Team, Biomedical Research Foundation of the Academy of Athens , Athens , Greece
4 Computer Engineering and Informatics Department, School of Engineering, University of Patras , Patras , Greece
Perez-Acle Tomas
Electronic publication date: 2013 Oct 22
Publication date: 2013
Volume: 1
Electronic Location ID: e185
Received 2013 Jun 7; Accepted 2013 Oct 2
Copyright: © 2013 Carvalho et al.
Copyright year: 2013
Copyright holder: Carvalho et al.
License: This is an open access article distributed under the terms of the Creative Commons Attribution License, which permits unrestricted use, distribution, and reproduction in any medium, provided the original author and source are credited.
License URL: https://creativecommons.org/licenses/by/3.0/

Keywords: Protein similarity search, Structural biology, Harmonic space, Electrostatic potentials, Drug design

Funding: European Social Fund Greek national funds FCT-Lisbon Grant no. SFRH/BPD/65993/2009 This study was supported by the European Union (European Social Fund—ESF), received Greek national funds through the Operational Program ‘Education and Lifelong Learning’ of the National Strategic Reference Framework (NSRF)—Research Funding Program: Shales, and the Investing in Knowledge Society through the European Social Fund. CSC is funded by the FCT-Lisbon, Grant no. SFRH/BPD/65993/2009. The funders had no role in study design, data collection and analysis, decision to publish, or preparation of the manuscript.

==============================
We developed a novel method based on the Fourier analysis of protein molecular surfaces to speed up the analysis of the vast structural data generated in the post-genomic era. This method computes the power spectrum of surfaces of the molecular electrostatic potential, whose three-dimensional coordinates have been either experimentally or theoretically determined. Thus we achieve a reduction of the initial three-dimensional information on the molecular surface to the one-dimensional information on pairs of points at a fixed scale apart. Consequently, the similarity search in our method is computationally less demanding and significantly faster than shape comparison methods. As proof of principle, we applied our method to a training set of viral proteins that are involved in major diseases such as Hepatitis C, Dengue fever, Yellow fever, Bovine viral diarrhea and West Nile fever. The training set contains proteins of four different protein families, as well as a mammalian representative enzyme. We found that the power spectrum successfully assigns a unique signature to each protein included in our training set, thus providing a direct probe of functional similarity among proteins. The results agree with established biological data from conventional structural biochemistry analyses.

Introduction

The spatial structure of proteins encodes information on their function which is essential for a successful drug design. In the post-genomic era, the search for functional similarities among proteins is based mostly on identity and/or similarity of genomic sequences rather than on their spatial structure. An approach that has been widely used is the application of self-organizing maps to the protein amino acid sequence in order to predict the protein shape and to infer the protein function (Kohonen, 1982; Andrade et al., 1997). This approach searches for local similarities in the amino acid sequence and is based on the assumption that the proteins have the same size and that the amino acid sequence is a determinant of the protein structure. However, there are many examples of proteins where sequence-based searches are insufficient to describe their biological function (Dobson et al., 2004). While self-organizing maps can classify proteins into families, they fail at predicting the structure. Since structure is more conserved than sequence, evolutionary relationships among proteins, protein structure-function predictions and comparative modelling should be based on structural information, rather than on primary amino acid or genomic sequence (Illergård, Ardell & Elofsson, 2009).

Other approaches have been developed that search for functional similarities using the complete three-dimensional information encoded in the spatial coordinates of all the atoms within the protein structure, which have been derived from X-ray or nuclear magnetic resonance experiments. In these approaches, the three-dimensional protein structure is modelled by a representation (or descriptor) based on topological characteristics or structure elements (see e.g., Venkatraman, Sael & Kihara, 2009 and references therein). Although the structure-based approaches are more informative on the function than the sequence-based ones, structure comparison methods are too slow and are thus rendered impractical to use in large-scale experiments and real-life applications (Kolodny, Koehl & Levitt, 2005; Mayr, Domingues & Lackner, 2007; Berbalk, Schwaiger & Lackner, 2009).

Other approaches use the protein solvent-accessible surface, since it is a stronger determinant of the protein function than sequence or structure (Via et al., 2000). Shape descriptors have been developed based on spatial symmetries, where the search for similarities consists of shape comparison (Kazhdan, Funkhouser & Rusinkiewicz, 2003; Ritchie, Kozakov & Vajda, 2008; Venkatraman, Chakravarthy & Kihara, 2009). However, surface comparison methods are computationally challenging, largely because they suffer from ambiguity in spatial orientation and require that the surfaces be aligned for an optimal matching (see, Via et al., 2000 and references therein).

Bioinformatics has become the new biomedical informatics bottleneck, as the cost of genome sequencing and the sheer quantity of genomic data has recently skyrocketed. It has been estimated that the unprocessed data generated per sequencing machine can be of order at least 30 Gbs per day, which can scale up by a significant factor in the case of mapped/processed data. There is a clear requirement for fast and efficient analysis of the entire genome/proteome sequencing data in the upcoming era of personalized medicine. Due to the continuous improvements in sequencing technologies and proteomic methodologies, the current scaling of available computing, storage and analysis throughput is far lower than the scaling of the data generation rate. The induced lag between the processing potential and the processing requirements already poses problems to researchers and companies in the bioinformatics field. Since it is impossible to constantly upgrade computer hardware to keep up with the increasing data production rate, the only feasible solution is to devise algorithms that can offer a competing processing scaling using the existing hardware at its full potential.

Here, we propose a new approach to search for functional similarities among proteins using their molecular surfaces (Vlachakis et al., 2012). Protein molecular surfaces are determinant of the protein biological activity, with different types of molecular surfaces encoding different information about the protein function. We choose to use surfaces of the molecular electrostatic potential due to the importance of the charge distribution in the protein-protein interactions. Protein-protein interactions are essential for cell signalling and cell function (Przytycka, Singh & Slonim, 2010; Berger-Wolf et al., 2010). These processes require a correct and fast molecular recognition in which interactions among electrostatic charges intervene. Disturbances in these processes are in the origin of almost every major disorder (Gire et al., 2012) and may lead to severe diseases such as cancer (Elcock et al., 1999; Sept, Elcock & McCammon, 1999; Wlodek, Shen & McCammon, 2000). Therefore the electrostatic potential distribution on the protein molecular surfaces is crucial to virtually all biological macromolecules involved in key biochemical pathways (Honig & Nicholls, 1995; Wong & Pollack, 2010; McCammon, 2009).

Once we calculate the molecular surface for a particular filter, we proceed to measure the signal off of the molecular surface. For our signal analysis, we propose a method based on the Fourier analysis of molecular surfaces. An advantage of Fourier analysis is that it most easily separates large from small scales. The signal at each point can be regarded as a realization of a distribution of fluctuations around an average value of the molecular surface (Vlachakis, Champeris-Tsaniras & Kossida, 2012; Kandil et al., 2009). Instead of measuring information on the individual points over the surface as shape descriptors do, we measure information on the correlations among the points, thus waiving the need that the surfaces be aligned. The simplest statistic is the two-point correlation function in Fourier space, which averages the signal over the whole volume and measures the variance in the distribution. Hence our approach transforms three-dimensional spatial data into one-dimensional frequency data.

The manuscript is organized as follows. First we present the selected proteins and how we synthesise the corresponding molecular surfaces. Then we describe our proposed method to extract functional information, based on the Fourier analysis of molecular surfaces and on a dimensionality reduction of the usable information. Then we present the results and discuss further improvements in the robustness of this method. Finally we outline an integrated solution for a functional similarity search among proteins, which progresses towards a dimensionality increase of the usable information and a reduction of the protein sample size.

Method

Selected proteins

For our training set, we selected four distinct protein families, which include twelve helicase proteins, six methyltransferase proteins, four polymerase proteins and four glycoproteins. These proteins are mainly viral components that are involved in major diseases such as Hepatitis C, Dengue fever, Yellow fever, Bovine viral diarrhea and West Nile fever. We use the Mouse kinase protein as a decoy, since it has a very different function from all the other proteins.

Helicases are responsible for the unwinding of double stranded DNA or RNA during viral replication. Polymerases are key enzymes that are used for copying the viral genetic material. Methyltransferases or methylases are transferase enzymes that are responsible for transferring methyl groups from a donor to an acceptor. Finally glycoproteins are used for molecular recognition by viruses. Protein treatments vary depending on the needs of each comparison chart. The main treatment is the default X-ray crystallography protein conformation as it is deposited in the RCSB database (Berman et al., 2000). The selected unedited proteins were the following. Among the helicases, we selected: (a) 1A1V and 8OHM of the Hepatitis C virus (HCV), (b) 1YMF, 1YKS and 2V80 of the Yellow fever virus (denoted by YF_1YMF, YF_1YKS and YF_2V80 respectively), and (c) 2JLU, 2BHR, 2BMF and 2JLQ of the Dengue fever virus (denoted by DEN_2JLU, DEN_2BHR, DEN_2BMF and DEN_2JLQ respectively). Among the polymerases, we selected 2CJQ, 2HCS and 2HCN of the West Nile fever virus (denoted by WN_2CJQ, WN_2HCS and WN_2HCN respectively). Among the methyltransferase, we selected 3EVA, 3EVB, 3EVC, 3EVD, 3EVE and 3EVF of the Yellow fever virus (denoted by YF_3EVA, YF_3EVB, YF_3EVC, YF_3EVD, YF_3EVE and YF_3EVF respectively). Among the glycoproteins, we selected 1NB7, 4DVN, 4DW4 and 4DW3 of the bovine diarrhea virus (denoted by BVDV_1NB7, BVDV_4DVN, BVDV_4DW4 and BVDV_4DW3 respectively).

In the Hepatitis C viral protein family, we considered two HCV helicase proteins, namely the HCV helicase strain A (the 1A1V entry, denoted by HCV_helicaseStrA) and the HCV helicase strain B (the 8OHM entry, denoted by HCV_helicaseStrB), whose three-dimensional coordinates were obtained from the RCSB database (Berman et al., 2000) of X-ray protein crystallography structures. Furthermore, we generated two simulations of the 1A1V protein crystal, namely the energy-minimized version (denoted by HCV_helicaseEM) and the molecular dynamics version (denoted by HCV_helicaseMD). Both the HCV_helicaseEM and the HCV_helicaseMD have been energetically minimized up to a gradient of 0.05. The HCV_helicaseMD has additionally been subject to a molecular dynamics simulation. We also established a homology model of the HCV helicase (denoted by HCV_helicaseHM) so that the in silico three-dimensional model of HCV was included in our training set. We also included an example of a non-helicase HCV viral protein, namely the 1NB7 structure of the HCV polymerase (the 1NB7 entry, denoted by HCV_polymerase).

Molecular surfaces of the selected proteins

Surfaces of the molecular electrostatic potential follow the nonlinear Poisson-Boltzmann equation (Konecny, Baker & McCammon, 2012; Unni et al., 2011). We solved numerically for the electrostatic potential using the finite-difference method as implemented in the APBS Software (Baker et al., 2001). The potential was calculated on a regular grid of size (65, 65, 65),5 with the grid-fill-by-solute parameter set to 80%. The dielectric constants of the solvent and the solute were set to 80.0 and 2.0, respectively. An ionic exclusion radius of 2.0 Å, a solvent radius of 1.4 Å and a solvent ionic strength of 0.145 M were applied. Default APBS charges and atomic radii were used.

Energy minimization (EM) removes any residual geometrical strain from each molecular system, whereas molecular dynamics (MD) simulates a periodic cytoplasm-like aqueous environment. Both EM and MD were performed with the Gromacs suite (Hess et al., 2008; Lindahl, Hess & van der Spoel, 2001; van der Spoel et al., 2005) through our previously developed graphical interface (Sellis, Vlachakis & Vlassi, 2009). Molecular dynamics took place in a periodic environment, which was subsequently solvated with the simple point-charge water model using the truncated octahedron box extending to 7 Å from each molecule. Partial charges were applied and the molecular systems neutralized with counter-ions as required. The temperature was set to 300 K, the pressure to 1 atm and the step size to 2 fs. The total time elapsed at each molecular complex run was 50 ns, using constant number of atoms, volume and temperature (NVT) throughout the calculation in a canonical environment. The results of the MD simulations were collected in a molecular trajectory database for further analysis.

The homology model was produced using Modeller (S˘ali & Blundell, 1993; Eswar et al., 2003) and was evaluated using the Procheck utility (Laskowski et al., 1996). This model was designed in order to include a computer modelled structure in our training set, which however shares high sequence identity with its template structure (approximately 90%).

Figure 1 Surfaces of the electrostatic molecular potential.

A: Hepatitis C helicase protein, B: Hepatitis C polymerase protein. The electrostatic potential is measured in eV, with range as shown in the corresponding colour bar.

The RCSB/PDB entries of the selected proteins are summarized in Table 1. In Fig. 1 we show surfaces of the electrostatic molecular potential for two HCV proteins, namely the helicase and the polymerase. The electrostatic potential is measured in eV. In these manuscript, we used the Connolly representation for the molecular surfaces (Connolly, 1983).

Table 1 Input data.

Protein families, protein PDB names and sizes of the corresponding molecular surfaces along the [x, y, z]-directions, measured in Å.

Family	Protein	Size = [lx, ly, lz]	
Helicase	HCV_helicaseStrA	[72.4, 64.8, 55.1]	
HCV_helicaseEM	[72.8, 65.1, 55.5]	
HCV_helicaseMD	[72.3, 65.5, 56.3]	
HCV_helicaseHM	[71.5, 65.7, 55.9]	
HCV_helicaseStrB	[61.9, 69.6, 61.7]	
DEN_2BHR	[93.5, 101.4, 76.8]	
DEN_2BMF	[84.4, 111.7, 106.0]	
DEN_2JLQ	[66.8, 69.6, 77.0]	
DEN_2JLU	[80.4, 95.0, 85.0]	
YF_1YKS	[62.2, 58.4, 67.8]	
YF_1YMF	[63.6, 58.2, 67.8]	
YF_2V8O	[49.2, 69.6, 67.6]	
Polymerase	HCV_polymerase	[59.0, 77.7, 65.0]	
BVDV_2CJQ	[74.4, 69.5, 64.5	
WN_2HCN	[78.5, 75.4, 61.1]	
WN_2HCS	[77.2, 75.2, 63.2]	
Methyltransferase	YF_3EVA	[43.9, 56.2, 62.7]	
YF_3EVB	[43.6, 56.4, 62.6]	
YF_3EVC	[43.8, 56.0, 62.2]	
YF_3EVD	[44.1, 55.5, 63.3]	
YF_3EVE	[44.6, 55.9, 65.2]	
YF_3EVF	[43.9, 56.1, 64.2]	
Glycoproteins	BVDV_4DVN	[46.2, 67.2, 68.0]	
BVDV_4DW3	[46.3, 68.5, 67.9]	
BVDV_4DW4	[46.8, 73.4, 67.6]	
Kinase	Mouse_kinase	[52.5, 69.0, 48.8]	

Power spectrum of molecular surfaces

Molecular surfaces contain information on a property of proteins along the three spatial dimensions. This property, in this case the values of the electrostatic potential, can be regarded as a field F(x) defined over points x on the surface. Functional information is encoded not only in the positions of the points but also in the correlations among points. The simplest correlation function that we can measure is that between pairs of points. The two-point correlation function ξ of the field F measures the convolution of the field over its complex conjugate (see e.g., Peacock, 1999) (1) ξ(r)≡F*(x)F(x+r)=1L3∫d3xF*(x)F(x+r).

The angle brackets indicate an averaging over the normalization volume, which here we take as the volume of the molecular surface, L3.

We assume that the field has a flat geometry and can be decomposed in a Fourier expansion of plane waves (2) F(x)=∑kFkexp[−ik⋅x],

where the wavenumber k relates with the frequency ν by k = 2π/ν. If the field has a curved geometry, then a Fourier expansion in spherical harmonics should be used instead. However, the difference between the two expansions only matters in scales of order the size of the molecular surface, which correspond to the smallest frequency. The smallest frequency is the zero-mode in the Fourier expansion and describes a global offset. The two-point correlation function becomes (3) ξ(r)=∑k∑k′Fk*Fk′exp[i(k−k′)⋅x]exp[−ik′⋅r].

Since the molecular surface is closed, the field is periodic within the size of the surface, which restricts the allowed wavenumbers to the harmonic boundary condition kn=(n2π/L)eˆk, where n∈{0, 1, …} is the order of the Fourier modes. As a consequence, all the cross terms with k′ ≠ k average to zero and the remaining sum is (4) ξ(r)=L2π3∫d3k|Fk|2exp[−ik⋅r].

Hence the correlation function is the Fourier transform of the power spectrum P(k) = |Fk|2. This relationship is known as the Wiener-Khinchin theorem. The power spectrum measures amplitude correlations among the modes and discards information on the phase. We proceed to compute the Fourier transform Fk of the molecular surface inferred over a regular grid. The Fourier-transformed surface measures the amplitude of the plane waves whose combination reproduces the information on the original surface. The frequencies of the plane waves range from the frequency corresponding to the extension of the surface (i.e., to n = 1), up to the Nyquist frequency corresponding to twice the bin size of the grid (i.e., to n = N/2, where N is the number of bins along a direction of the grid). The size of the molecular surfaces ranges between 5 and 7 nm (Table 1). The smallest spatial scale of biological interest is the size of a typical cluster of aminoacids, which is of order xball ∼ 0.3 nm. We choose this spatial scale for the size of the grid, so that the largest frequency scale that can be probed is of order kball ∼ 10 nm−1.

Furthermore, we assume that the field is isotropic, i.e., that it does not have a preferential direction, so that the power spectrum depends only on the distance between each pair of points. (See Fig. 2A for an illustration.) By assuming isotropy, we are discarding information on the direction. We proceed to take the ensemble average of P(k) so that the power at the mode k is the sum of the power at all the points on a sphere of radius k from the zero-mode, resulting in a one-dimensional function P(k). In this way, we collapse the information on the three-dimensional field over the molecular surface onto a one-dimensional power spectrum over the wavenumbers of the Fourier-transformed molecular surface.

Figure 2 Schematic representation of point configurations for correlations in harmonic space.

A: The configuration of the two-point correlation function contains one free parameter, k12, which is the distance in harmonic space between the two points P1 and P2. B: The configuration of the three-point correlation function contains two free parameters, e.g., k12 and k23, describing the distances in harmonic space respectively between P1 and P2, and between P2 and P3. The third parameter k13 is related to the former two by the triangle condition k12 + k23 + k31 = 0.

For a given k, we are sampling a distribution, which we assume to be Gaussian with mean value Fk and variance |Fk|2=P(k), from which the Fourier coefficients Fk are drawn. Hence there is a fundamental uncertainty about the underlying variance, which depends on the number of coefficients sampled at a given k. Since the number of k’s on a sphere of radius k scales as k2 and for any real field it holds that F−k = Fk*, where the asterisk stands for the complex conjugate, then the uncertainty scales as ΔP(k)/P(k)=2/k2.

Figure 3 Power spectrum of the molecular surfaces of the selected proteins.

Power spectra of the corresponding white-noise molecular surfaces of some helicase and polymerase proteins. The values of k are measured in nm−1.

Figure 4 Power spectrum of the molecular surfaces of the selected HCV helicase proteins.

Power spectra of the molecular surfaces divided by the power spectra of the corresponding white-noise molecular surfaces. The symbols depict the power spectra and the error bars depict the error associated with the measurement. The values of k are measured in nm−1.

Figure 5 Power spectrum of the molecular surfaces of the selected Dengue virus helicase proteins.

Power spectra of the molecular surfaces divided by the power spectra of the corresponding white-noise molecular surfaces. The symbols depict the power spectra and the error bars depict the error associated with the measurement. The values of k are measured in nm−1.

Figure 6 Power spectrum of the molecular surfaces of the selected Yellow fever virus helicase proteins.

Power spectra of the molecular surfaces divided by the power spectra of the corresponding white-noise molecular surfaces. The symbols depict the power spectra and the error bars depict the error associated with the measurement. The values of k are measured in nm−1.

Figure 7 Power spectrum of the molecular surfaces of the selected polymerase proteins.

Power spectra of the molecular surfaces divided by the power spectra of the corresponding white-noise molecular surfaces. The symbols depict the power spectra and the error bars depict the error associated with the measurement. The values of k are measured in nm−1.

Figure 8 Power spectrum of the molecular surfaces of the selected methyltransferase proteins.

Power spectra of the molecular surfaces divided by the power spectra of the corresponding white-noise molecular surfaces. The symbols depict the power spectra and the error bars depict the error associated with the measurement. The values of k are measured in nm−1.

Figure 9 Power spectrum of the molecular surfaces of the selected glycoproteins proteins.

Power spectra of the molecular surfaces divided by the power spectra of the corresponding white-noise molecular surfaces. The symbols depict the power spectra and the error bars depict the error associated with the measurement. The values of k are measured in nm−1.

Results

Power spectrum of the molecular surfaces of the selected proteins

To test our method, we used the protein simulations described above for a training set, containing four different protein families. For each molecular surface of the electrostatic potential, we computed its power spectrum and the corresponding white-noise power spectrum. The white-noise power spectrum was computed from a surface synthesised as a Gaussian distribution N(0, 1) times the mean value of the corresponding molecular surface. We observe that the power spectra of all molecular surfaces have comparable magnitudes, stabilizing around 10−6 for sufficiently large k (not shown), whereas the white-noise power spectra have magnitudes that range from 10−11 to 10−7 (Fig. 3). This range is populated by the HCV_polymerase at the top, followed by the HCV_helicaseStrB and HCV_helicaseHM in the intermediary range, and finally the 1A1Vs helicase HCV_helicaseStrA and its models HCV_helicaseEM and HCV_helicaseMD at the bottom. Hence the information derived from the mean value alone, assuming an underlining Gaussian distribution, suggests a coarse clustering of the proteins in helicases, polymerases and a mixed cluster containing helicases and non-helicases.

For an easier comparison of the results, we divided the power spectra of molecular surfaces by the mean of the corresponding white-noise power spectra (Figs. 4–9). For each molecular surface, the power at each k is one realization of a distribution, hence the power spectrum is noisy. This noise was estimated by the uncertainty of the Fourier coefficients at each k, given by ΔP(k)=P(k)2/k2, which we used to compute the error bars. We also included the power spectrum of Mouse_kinase in all plots, which shows a nearly flat spectrum punctuated by irregular peaks.

First we analyse the HCV helicase protein set which illustrates how our method performs at distinguishing different treatments and strains of the same protein. We plotted the power spectra of the HCV helicase proteins in Fig. 4. We observe that the power spectra of HCV_helicaseStrA, HCV_helicaseEM, HCV_helicaseMD and HCV_helicaseStrB exhibit a similar pattern up to k ≈ 10 nm−1 compatible with that of HCV_helicaseHM.

A further inspection reveals details that distinguish among the helicases. In particular, we observe that the power spectra of the models HCV_helicaseEM and HCV_helicaseMD exhibit very similar patterns of peaks attesting to their similar binding state. Although HCV_helicaseStrA is in a different binding state, its power spectrum exhibits the same level of similarities with both HCV models, with an anticipated HCV-like grouping of peaks specific to our data. For k > 1 nm−1, these three helicase proteins exhibit three strong peaks at k ≈ 2.3, 4.6, 7.3 nm−1. From the distance between peaks, we infer an average wavelength of λ ≈ 2.5 nm. The power spectrum of HCV_helicaseHM follows the same pattern as that of HCV_helicaseStrA shifted to smaller k with a varying relative phase which most of the time is close to π, with strong peaks at k ≈ 1.8, 3.6 nm−1. In comparison with the HCV models, the power spectrum of HCV_helicaseStrB exhibits differences in the position of the peaks (found at k ≈ 2.7, 3.6, 5.5 nm−1) and in their amplitude ratios, which attest to the different treatment in HCV_helicaseStrB from that in the HCV models. As k increases, we observe a gradual damping of the power of the helicases and an emerging tail reminiscent of shot noise in a Poisson power spectrum, more prominent in HCV_helicaseHM and HCV_helicaseStrB, which indicates the damping of the fluctuations about the mean value and thus the vanishing of the structural signal. This damping is most visible for k > 6 nm−1. This agrees with the observation above that sets the upper limit of k to the size of a typical cluster of aminoacids and hence sets the minimum distance below which correlations are not of biological interest nor can be reliably probed by X-ray/NMR experiments.

We now proceed to analyse the remaining helicase strains, which illustrates how our method performs at distinguishing strains of the same family. We plotted the power spectra of the Dengue virus (DEN) helicase proteins in Fig. 5 and the Yellow fever virus (YF) helicase proteins in Fig. 6.

The power spectra of both the YF helicase proteins and the DEN helicase proteins exhibit a similar pattern with a varying relative phase among the proteins of each strain, with the difference between the two strains being in the typical wavelength and amplitude. In particular, we observe that the DEN helicases have an underlying flat spectrum punctuate by peaks at k ≈ 2.0, 3.8, 5.5 nm−1 (DEN_2BHR), k ≈ 3.0, 5.0, 6.0 nm−1 (DN_2JLQ) and k ≈ 3.5, 5.0, 6.5 nm−1 (DN_2JLU), that yield an average λ ≈ 4.2 nm. The DEN_2BMF shows a different pattern characterized by a decreasing power law up to k ≈ 5, with superposed peaks k ≈ 5.5, 7.0, 8.5 nm−1. In contrast, the YF helicases have a nearly flat spectrum punctuated by small peaks at k ≈ 1.0, 3.5, 5.5 nm−1 (YF_1YKS) and k ≈ 1.5, 5.5 nm−1 (YF_1YMF), that yield an average λ ≈ 2.0 nm. The YF_2V80 shows a nearly flat spectrum with barely no peaks, indicating a predominantly isotropic distribution of power. These families have a similar pattern with the HCV helicases but the features have smaller amplitudes. The global pattern attests to the fact that these proteins are also helicases and have the same treatment as the HCV, whereas the differences in amplitude attest to the fact that are of different strains.

We now proceed to analyse the non-helicase families, which illustrates how our method performs at distinguishing protein families.

We plotted the power spectra of the polymerase proteins in Fig. 7. We observe that all the polimerases have the same pattern characterized by an underlying decreasing power law with superposed peaks. In particular, the West Nile strains have the same pattern at all scales and a peak at k ≈ 8 nm−1, i.e., close to the smallest scale accessible. The BVDV strain has a very similar power law behaviour to the WN strains but is punctuated by regular peaks namely at k ≈ 1.5, 3.0, 4.5, 6.0, 8.0 nm−1, corresponding to an average λ ≈ 4 nm. The HCV polymerase has the steepest decreasing power law behaviour and peaks at k ≈ 5.5, 7.5 nm−1.

We plotted the power spectra of the methyltransferase proteins in Fig. 8. We observe that all the YF methyltransferase have similar patterns characterized by a nearly flat, featureless power spectrum punctuated by irregular low-amplitude peaks.

Finally, we plotted the power spectra of the glycoproteins in Fig. 9. We observe that all the BVDV glycoproteins have the same pattern characterized by an underlying a convex quadratic function with superposed peaks. In particular both the strains 4DVN and 4DW3 have a single peak at k ≈ 7.5 nm−1, whereas the 4DW4 have peaks at k ≈ 2.0, 4.5, 6.0, 8.0 nm−1, corresponding to an average λ ≈ 3.3 nm.

Power spectrum of a dynamical simulation

To further test our method, we used the 1A1V template energetically minimized up to a gradient of 10−5 to generate ten dynamical realizations captured in ten time frames separated by 10 ps. We then energetically minimized the tenth frame up to a gradient of 10−5 (Vangelatos et al., 2009; Sellis et al., 2012; Vlachakis et al., 2013). We computed the power spectrum of each frame, generated the corresponding white noise surface and plotted the results in Fig. 10.

The purpose of this test is to show how our method behaves when applied to controlled simulations. We observe that there is no significant difference among the different frames. This observation supports the fact that the surfaces do not change over time after energy minimization (EM). Also we observe that the two simulations energetically minimized up to a gradient 10−5 are in phase, whereas the simulation with up to a gradient 5 × 10−2 is visibly out of phase with the former. This observation supports the fact that there is a difference between crude and fine EM.

Figure 10 Power spectra of the molecular surfaces of the HCV_helicaseEM after being subject to molecular dynamics simulations for 100 ps.

Power spectra of the molecular surfaces divided by the power spectra of the corresponding white-noise molecular surfaces. The symbols depict the power spectra and the error bars depict the error associated with the measurement. The values of k are measured in nm−1.

Conclusions

We presented a new method based on the Fourier analysis of protein molecular surfaces to extract functional information on proteins. For a selected set of proteins of HCV with different structural features, we first produced surfaces of the molecular electrostatic potential, as well as the corresponding white-noise surfaces, and then computed their two-point correlation function in harmonic space (the power spectrum). We found that this method can distinguish different functional protein groups. More specifically, in this manuscript we established a helicase, a polymerase, a methyltransferase and a glycoprotein group. We also tested this method on dynamical simulations after energy minimization.

An immediate extension of this work is the application of this method to isolated structural subunits that form larger structures within proteins. Similarly sized subunits will have a strong signal in the same frequency range, which will add up in the protein power spectrum. Hence, we must first measure the contribution of each subunit separately and produce a catalogue of subunit signatures, so we can distinguish them in the combined signal when running similarity searches.

By reducing the initial three-dimensional information on the molecular surface to the one-dimensional information on pairs of points at a fixed scale apart, this method allows for a fast similarity search. Further refinements in the similarity search will require methods that use information from higher-order correlation functions, such as the correlation among three points at a fixed triangular configuration or its Fourier-transformed (the bispectrum). (See Fig. 2 right panel for an illustration.) The bispectrum measures phase correlations among the modes and thus deviations from a Gaussian distribution.

Our ultimate goal is to integrate higher-order correlations and to apply the resulting method to the RCSB database so as to provide the biopharmaceutical and structural research communities with a novel and easily searchable reference without the three-dimensional information compromising the speed of the calculation. This method aims to coalesce techniques, which have been extensively tested and used in other fields such as cosmology, into a fast and robust pipeline for the analysis and processing of very large, three-dimensional biological datasets in an effort to speed up protein similarity searches.

Additional Information and Declarations

Competing Interests

Author Contributions

5 The size of the grid was kept small in order to speed up the calculation and reduce the computational load. It was tested to be suitable for this study, as higher detail would not change the surface much, while it would increase the computational load significantly.

The authors declare there are no competing interests.

C. Sofia Carvalho, Dimitrios Vlachakis, Georgia Tsiliki, Vasileios Megalooikonomou and Sophia Kossida conceived and designed the experiments, performed the experiments, analyzed the data, contributed reagents/materials/analysis tools, wrote the paper.

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
