# Peer review of "Protein signatures using electrostatic molecular surfaces in harmonic space"

_PeerJ, doi:10.7717/peerj.185_

## Round 0.1 · original submission · Major Revisions

· Academic Editor

Major Revisions

Dear authors,
Despite that I have made extensive efforts to find additional reviewers for your article entitled “Protein signatures using electrostatic molecular surfaces in harmonic space”, I haven't succeeded. In order to speed up the evaluation process of your article and considering my expertise in structure-function relationships in protein structure, I have acted as the second reviewer. As a whole, my overall decision is that your paper should be accepted for publication at PeerJ once all issues raised by the review process are properly addressed. Please find my review below (and you will find the Comments of Reviewer 1 below that).

Basic Reporting:
The article entitled “Protein signatures using electrostatic molecular surfaces in harmonic space” from authors Carvalho, Vlachakis, Tsiliki, Vasileios, Megalooikonomou and Kossida, presents a novel approach to use the fourier spectrum computed from the electrostatic potential derived from protein surfaces as a comparative value among protein structures. According to the authors, this method reduces the 3D information available at the protein surface to 1D information where a specific distance metric in the harmonic space may be used to compare protein structures.

Experimental design:
By relying on the Wiener-Khinchin theorem, the authors of this paper correlate different points of the electrostatic field computed from an APBS calculation by applying a Fourier transform of the power spectrum inferred over a regular grid over the protein surface. The authors evaluated their method by using a training set composed by 3 helicase proteins from HCV, including 2 crystal structures and 1 comparative model. They also included as decoys 1 HCV polymerase and 1 cAMP-dependant kinase.

Validity of the findings:
Minor concerns
- Despite that the authors succeeded presenting a method to compare protein structures relying on 1D information, it is unclear how this method can be used to extract functional information from proteins. Moreover, one of the authors’ claims is that this method allows "a fast similarity search". Though comparing 1D information is faster than 3D, the authors do not consider that their method requires a previous calculation of the electrostatic potential of every protein structure. Is the time needed to compute the electrostatic potential being considered as part of the method? On the other hand, in order to better evaluate the performance of their method, the authors should compare their results with well-established methods such as VASP and others.
- Notoriously, the elegant approach by which the authors use the Fourier transform of the power spectrum computed on the electrostatic potential of protein surfaces, heavily contrast with the use of a simple “distance metric” by which they compare the power spectrum of different proteins. The inclusion of a more robust method such as SVM or Kohonen maps to compare proteins, could improve their findings.
- In order to simplify their calculations, the authors have assumed that electrostatic field is isotropic so that the power spectrum depends only on the distance between each pair of points. However, it is well known that in the surface of proteins is where the side chain of charged residues is actually located. At physiological pH, many residues can be charged forming dipoles that are sensitive to the charge flow. Therefore, It is unclear how this simplification can be used to account for changes in electrostatic potential due to flow of charges as in the case of MD simulations.

Major concerns:
- Overall, the major concern behind this interesting work is the scarcity of the protein structure database that was used as test set. Despite that the inclusion of decoys is an elegant way to approximate to the real values of precision and recall, the authors should make a compelling effort to increase the size of both the training and the test set used to evaluate this study. They should also improve the statistics behind the evaluation of their method by taking a look at the actual predictive value (precision and recall).

Reviewer 1 ·

Basic reporting

The contribution report a new method of analysis of protein molecular surface based on Fourier analysis. The authors calculate a power spectrum of surfaces of the molecular electrostatic potential, to compare with other molecules in a structural database. This method allows a reduction of the three-dimensional information to one-dimensional information. They use a training set of the Hepatitis C viral proteins.
This approach shows a contribution to speed-up the comparison and classification of proteins. The method is mathematically correct and allows the calculation a power spectrum that can easily be comparing between different molecules.

Experimental design

The training set is small to can show the real meaningful of the methods, in fact the author use only to structure, Helicase A and Helicase B, all the other structures were derivatives of one of them, so they present only small differences with the original, and in this case are not candidate to show the advantages of the methods. In fact, one of them present a power spectrum different to the non related protein (Polymerase) but the other present a power spectrum similar to the other non related protein ( kinase).

Validity of the findings

The findings are based in a trainning set not robust, I will suggest use a bigger and robust training set, to show the real effect of the methods.

Additional comments

This a good contribution that need inclrese the training set and demeostrate tha capability of the powwr spectrum to differenciate non related protein.

---

## Round 0.2 · accepted · Accept

· Academic Editor

Accept

Dear authors,
Thank you very much for your efforts to include the comments and solicitations of the reviewing panel. It is now our pleasure to publish your paper in accordance to the previous decision. Thanks a lot for considering PeerJ to be the showcase of this scientific contribution. Looking forward for your future contributions in this field.